# Diagnostic performance of GENEDIA W and ActiveXpress+ COVID-19 antigens tests among symptomatic individuals in Peru and The United Kingdom

**Sandra Palomino-Padilla[1], Lorna Finch[2], Margaretha de Vos[3], Helen Savage[2], Luz Villa-Castillo[1], Gail Hayward[4], Eloïse Cook[5], LSTM diagnostics group[2¶], UTB-IMTAvH group[1¶], CONDOR steering group[¶], Camille Escadafal[3], Richard Body[5], Emily R. Adams[2], Cesar Ugarte-Gil[1‡]\*, Ana I. Cubas-Atienzar[2‡]\***

1 Liverpool School of Tropical Medicine, Centre for Drugs and Diagnostics, Liverpool, United Kingdom, 2 Unidad de Investigación de Tuberculosis del Instituto de Medicina Tropical Alexander von Humboldt, Universidad Peruana Cayetano Heredia, Lima, Peru, 3 FIND, Geneva, Switzerland, 4 Oxford University, Oxford, United Kingdom, 5 Manchester University NHS Foundation Trust, Manchester, United Kingdom

‡ CUG and AICA are joint last senior authors.
¶ Membership of the LSTM diagnostics group, UTB-IMTAvH group and CONDOR steering group are provided in the Acknowledgments.
\* Ana.CubasAtienzar@lstmed.ac.uk (AICA); Cesar.Ugarte@upch.pe (CUG)

**Data Availability Statement:** All relevant data are available within the paper and its Supporting Information files.

## Abstract

### Objectives

In order to generate independent performance data regarding accuracy of COVID-19 antigen-based rapid diagnostic tests (Ag-RDTs), prospective diagnostic evaluation studies across multiple sites are required to evaluate their performance in different clinical settings. This report describes the clinical evaluation the GENEDIA W COVID-19 Ag Device (Green Cross Medical Science Corp., Chungbuk, Korea) and the ActiveXpress+ COVID-19 Complete Testing Kit (Edinburgh Genetics Ltd, UK), in two testing sites Peru and the United Kingdom.

### Methods

Nasopharyngeal swabs collected from 456 symptomatic patients at primary points of care in Lima, Peru and 610 symptomatic participants at a COVID-19 Drive-Through testing site in Liverpool, England were analyzed by Ag-RDT and compared to RT-PCR. Analytical evaluation of both Ag-RDTs was assessed using serial dilutions of direct culture supernatant of a clinical SARS-CoV-2 isolate from the B.1.1.7 lineage.

### Results

For GENEDIA brand, the values of overall sensitivity and specificity were 60.4% [95% CI 52.4–67.9%], and 99.2% [95% CI 97.6–99.7%] respectively; and for Active Xpress+ the overall values of sensitivity and specificity were 66.2% [95% CI 54.0–76.5%], and 99.6% [95% CI 97.9–99.9%] respectively. The analytical limit of detection was determined at 5.0 x

**Funding:** This work was funded as part of FIND's work as co-convener of the diagnostics pillar of the Access to COVID-19 Tools (ACT) Accelerator, including support from Unitaid [grant number: 2019-32-FIND MDR], the governments of the Netherlands [grant number: MINBUZA-2020.961444] and from UK Department for International Development [grant number 300341-102]. The FALCON study was funded by the National Institute for Health Research, COV0051, Richard Body; Asthma UK, and the British Lung Foundation. The funders had no role in the study design, data collection, analysis, the decision to publish, or the preparation of the manuscript.

**Competing interests:** The authors have declared that no competing interests exist.

$10^2$ pfu/ml what equals to approximately $1.0 \times 10^4$ gcn/ml for both Ag-RDTs. The UK cohort had lower median Ct values compared to that of Peru during both evaluations. When split by Ct, both Ag-RDTs had optimum sensitivities at Ct<20 (in Peru; 95% [95% CI 76.4–99.1%] and 100.0% [95% CI 74.1–100.0%] and in the UK; 59.2% [95% CI 44.2–73.0%] and 100.0% [95% CI 15.8–100.0%], for the GENDIA and the ActiveXpress+, respectively).

## Conclusions

Whilst the overall clinical sensitivity of the Genedia did not meet WHO minimum performance requirements for rapid immunoassays in either cohort, the ActiveXpress+ did so for the small UK cohort. This study illustrates comparative performance of Ag-RDTs across two global settings and considers the different approaches in evaluation methods.

## 1. Introduction

The impact of the Severe Acute Respiratory Syndrome Coronavirus-2 (SARS-CoV-2), pandemic on healthcare has no precedents, stretching healthcare systems all around the world [1, 2]. The development of rapid, and easy-to-perform diagnostic tools has been a priority, and various rapid diagnostic tests for the detection of SARS CoV-2 antigens (Ag-RDTs) have been developed since the pandemic and are currently available on the market. FIND lists 206 SARS CoV-2 Ag-RDTs that are currently marketed or in development (data accessed November 2021) [3]. Despite the myriad of Ag-RDTs that are currently commercially available, at the time of this study there are only a few studies that evaluate their clinical accuracy directly from patients' samples.

In Peru, at the beginning of the pandemic, only a few laboratories had the capacity to test by RT-PCR for SARS-CoV-2 as this method requires adequate infrastructure, equipment, and well-trained professionals. In addition to these factors, the high global demand for RT-PCR reagents led to shortages of the tools needed to perform this test, as a result, Peru was considered one of the countries with the lowest diagnostic testing capacity for COVID-19 in Latin America [4], with approximately 500 RT-PCR tests per day performed in a country with a population of 32.971.846 individuals. Posteriorly, the diagnostic capacity increased up to ~10,000 tests per day but despite efforts to obtain the majority of molecular test results in the shortest period of time, the number of cases of COVID-19 was rapidly increasing and the RT-PCR testing capacity remained insufficient to test all the population who needed it. In this context, it became crucial to explore other diagnostic options, particularly as antibody based-RDTs were misused as a diagnostic tool for acute infection [5, 6]. At the time of study enrollment, Peru was facing the beginning of the second wave [7]which allowed to evaluate the performance of Ag-RDTs in a real-life context, and test its potential as a rapid diagnostic tool, to facilitate epidemiological surveillance and contact tracing in low-resource settings.

In the UK, following the WHO and European Commission emphasis on testing to halt the spread of SARS-CoV-2, capacity for mass testing significantly increased towards ~500,000 RT-PCR tests per day by November 2020, with a laboratory-based testing capacity reaching over 800,000 tests a day in January 2021 [8]. Through the UK governments' COVID-19 testing strategy, tests are available through NHS facilities, mobile testing units and satellite centres and includes 50 Regional Drive-Through COVID-19 Test Sites across the UK [8, 9]. Trials for Ag-RDTs using the Innova SARS-CoV-2 Antigen Rapid Qualitative Test (Innova Medical group, UK) were introduced across UK hospital staff, Armed forces and School cohorts, with a

city-wide mass testing implementation in Liverpool [10, 11]. Whilst these studies provide an insight into the accuracy of such tests in the UK, challenges in testing capacity and Ag-RDT availability as well as differences in SARS-CoV-2 variant circulation and prevalence in other countries are likely to impact the agreement of findings and adoption of particular Ag-RDTs across global settings. Prospective diagnostic evaluation studies across multiple, independent sites are required to determine the accuracy of COVID-19 Ag-RDTs that are affordable and accessible to LMICs, in order to promote equitable access to reliable diagnosis and support containment of SARS-CoV-2 around the world.

This study evaluates the clinical performance of two brands of Ag-RDTs commercially available at the time of this study: GENEDIA W COVID-19 (Green Cross Medical Science, Korea) referred to as GENEDIA and the ActiveXpress+ COVID-19 Complete Testing Kit (Edinburgh Genetics Ltd, UK) referred to as the ActiveXpress+. The evaluated ActiveXpress + version is no longer commercialized. Both Ag-RDTs were evaluated against SARS-CoV-2 RT-qPCR testing in Peru and the UK in different settings: two primary health centers and one hospital in Lima (Peru) and a National COVID-19 Drive-Through Testing Centre publicly open to symptomatic individuals located in Liverpool (UK).

## 2. Methods

### 2.1 Study design, settings and participants

This was a prospective evaluation of consecutive participants enrolled in two different settings:

In Peru, adults presenting with symptoms of COVID-19 were invited to participate in the study between January and March of 2021. Recruitment took place at two primary health centers and one hospital: Max Arias Health Center, Medalla Milagrosa Health Center and Huaycán Hospital, all located in Lima, Peru. Ethical approval was obtained from the Ethics Committee of Universitad Peruana Cayetano Heredia (SIDISI:202734) and was registered at the Peruvian National Institute of Health repository (PRISA) with the number EI00000001410.

Following all the safety precautions, trained clinical research nurses obtained the written informed consent form and started to collect two nasopharyngeal (NP) swab samples from each participant, one nostril using commercially available swabs and the other one using manufacturers swabs following IFU of Ag test. The first swab was used for the Ag-RDT that was performed immediately after collection of samples at the clinical site, where results were reading after 15–20 minutes following the instructions for use (IFU) by manufacturer. While the second NP swab was placed in universal transport media (UTM) Puritan® UniTranz-RT Transport Systems (Puritan Medical Products, USA) and transported in coolers to the Laboratory of the Institute of Tropical Medicine Alexander Von Humboldt at the University Peruana Cayetano Heredia for the reference SARS-CoV-2 RT–qPCR testing. RNA extraction was performed using the Qiagen viral RNA extraction Kit (Qiagen, Germany). To amplify viral RNA, 8 μL was run on the CFX96 Touch Real-Time PCR Detection System (BIORAD, USA) using the Norgen's 2019-nCoV TaqMan RT-PCR Kit (Norgen Biotek Corp., Canada). Positive and negative controls were included in each run, as well as an internal control.

In the UK, adults presenting with symptoms of COVID-19 at a national community testing facility, the Liverpool John Lennon Airport Drive-Through COVID-19 test centre, were asked to participate in the study and provide a tacit consent. Participants were recruited between January and May of 2021 under the Facilitating Accelerated COVID-19 Diagnostics (FALCON) study. Ethical approval was obtained from the National Research Ethics Service and the Health Research Authority (IRAS ID:28422, clinical trial ID: NCT04408170).

Paired swabs were taken systematically by a trained healthcare professional, NP swab samples were collected to perform the Ag-RDTs and followed by combined throat and nose (TN) swab samples in UTM (Copan Diagnostics Inc, Italy) used for the reference RT-qPCR test, following the national standard of care. Swab samples were obtained by healthcare professionals and transported with coolers to the Liverpool School of Tropical Medicine (LSTM). Recommended swabs for each Ag test were processed by trained laboratory researchers according to the Ag-RDT IFU. Both Ag-RDTs were performed upon arrival, with a time delay of 1–3 hours, on the dry NP swabs, while TN swabs in UTM were aliquoted and stored at -80˚C until RNA extraction. RNA was extracted using the QIAamp® 96 Virus QIAcube® HT kit (Qiagen, Germany) on the QIAcube® (Qiagen, Germany) and screened using TaqPath COVID-19 (ThermoFisher, UK) on the QuantStudio 5$^{TM}$ thermocycler (ThermoFisher, UK). Positive and negative controls were included in each run, as well as an internal control according to IFU.

## 2.2 Analytical sensitivity

A SARS-CoV-2 strain (202,012/01) from the B.1.1.7 lineage (Genbank accession number: MW980115), was used to investigate the limit of detection (LOD) of GENEDIA and the ActiveXpress+. Frozen aliquots of the third passage of the virus were quantified via plaque assay. For the determination of the LOD, a fresh aliquot was serially diluted from $1.0 \times 10^6$ plaque forming units (pfu)/ml to $1.0 \times 10^2$ pfu/ml. Each dilution was tested in triplicate. Two-fold dilutions were made below the ten-fold LOD dilution to confirm the lowest LOD. Culture media was used as negative control.

Viral RNA was extracted from each dilution using QIAmp Viral RNA mini kit (Qiagen, Germany) according to the manufacturer's instructions, and quantified using Genesig RT-PCR (Primer Design, UK). Genome copy number/ml (gcn/ml) were calculated as previously described [12].

## 2.3 Statistical analysis

The sensitivity and specificity, with 95% confidence intervals (CIs) of the GENEDIA and ActiveXpress+ devices were calculated based on the results of the reference method by RT-qPCR assay. Statistical analyses were performed using R scripts and GraphPad Prism 9.1.0 (GraphPad Software, Inc, California). The 95% confidence interval (CI) for the sensitivity and specificity was calculated using Wilson's method [13]. Fisher's exact and chi-squared tests were used to determine non-random associations between categorical variables.

# 3. Results

## 3.1 Clinical evaluation

The demographics of both the Peruvian and UK study cohorts are shown in Table 1. In Peru the median days from onset of symptoms was 6 days with no vaccination in Peru during recruitment. In the UK the median days from onset of symptoms was 2 days and a vaccination level of 21.15% was found across the combined cohort for GENEDIA and ActiveXpress+ evaluation. The clinical sensitivity of both Ag-RDTs were very heterogeneous between cohorts.

In Peru, of the 228 participants recruited, 108 samples were tested with Genedia and 120 with ActiveXpress+ Ag-RDT. Specifically, for GENEDIA W COVID-19 Ag Device evaluation, fifty-four of 108 (50%) samples were positive by RT-PCR (See Table 2). Thirty-nine of the RT-PCR positive samples (72.22%) were Ag-RDT positive, while the remaining 15 (27.78%) were Ag-RDT negative. Of the 54 RT-PCR negative specimens, only one was antigen-positive. The sensitivity and specificity for GENEDIA W COVID-19 Ag-RDT test based on RT-PCR

**Table 1. Demographics of Ag-RDT clinical evaluation cohorts for Peru and United Kingdom.**

| Country | Peru | | United Kingdom | |
|---|---|---|---|---|
| Characteristic | GENEDIA | ActiveXpress+ | GENEDIA | ActiveXpress+ |
| Age [mean (min-max), N] | 37.8 (18–80), 108 | 39.1 (18–68), 120 | 42.6 (18–83), 403 | 40.9 (18–77), 221 |
| Gender [%F, (n/N)] | 48.1%; (53/108) | 67.7%, (74/120) | 56.7% (221/388) | 61.8% (131/212) |
| Days from symptom onset [median (Q1-Q3); N] | 5 (3.8–7), 108 | 4 (3–6), 120 | 1 (1–3), 388 | 2 (1–3), 203 |
| Days < 0–3 (n, %) | 27, 25.0% | 31, 26.0% | 316, 81.4% | 161, 79.3% |
| Days 4–7 (n, %) | 66, 61.0% | 74, 62.0% | 48, 12.4% | 32, 15.8% |
| Days 8+ (n, %) | 15, 14.0% | 15, 12.0% | 24, 6.2% | 10, 4.9% |
| Vaccinated (n, %) | 0% | 0% | 40, 10.0% | 89, 42.2% |
| Not vaccinated (n, %) | 100% | 100% | 224, 56.1% | 117, 55.5% |
| Vaccination not disclosed (n, %) | 0% | 0% | 135, 33.8% | 5, 2.4% |

were 72.2% [95% CI 59.1% to 82,4%] and 98.1% [95% CI 90.2 to 99.7%], respectively. In this group the median days from symptom onset was 5 days [Q1-Q3 3.8–7] and median for PCR Ct values was 22.5 [Q1-Q3 18.1–27.1]. In the ActiveXpress+ COVID-19 evaluation, fifty-four of 120 (45%) specimens were positive by RT-PCR (See Table 3). Thirty-three of the RT-PCR positive samples (61.1%) were Ag-RDT positive; the 21 remaining results were all Ag-RDT negative. None of the 21 RT-PCR negative samples were Ag-RDT positive. The sensitivity and specificity for ActiveXpress+ COVID-19 Ag test based on RT-PCR were 61% [95% CI 47.8% to 73%] and 100% [CI 95% 94.5% to 100%], respectively. In this group the median days from symptom onset was 4 days [Q1-Q3 3–6] and median for PCR Ct values was 24.1 [Q1-Q3 20.3–31.2].

In the UK, of the 615 participants recruited, 399 samples were tested with Genedia and 212 with ActiveXpress+ Ag-RDT, four participants from Genedia group and one from ActiveXpress+ were PCR undetermined and excluded from all analysis due to not have paired data. Ninety-five (23.8%) of 399 specimens collected during the enrolment period of the GENEDIA evaluation were positive for COVID-19 by RT-qPCR (see Table 2). Fifty-one of the RT-PCR-positive samples (53.7%) were Ag-RDT positive, while the remaining were Ag-RDT negative. Of the 304 RT-qPCR negative specimens, only 2 were Ag-RDT positive. This UK evaluation of the GENEDIA Ag-RDT showed a sensitivity and specificity of 53.7% [95% CI 43.1–64.0%] and

**Table 2. Results and clinical sensitivity and specificity of the GENEDIA W COVID-19 Ag Device based on COVID-19 RT-qPCR result in the UK and Peru.**

| Results of GENEDIA W COVID-19 Ag Device | Peru | | | United Kingdom | | |
|---|---|---|---|---|---|---|
| | Confirmed by RT-qPCR | | | | | |
| | Positive | Negative | Total | Positive | Negative | Total |
| Positive | 39 | 1 | 40 | 51 | 2 | 53 |
| Negative | 15 | 53 | 68 | 44 | 302 | 346 |
| Total | 54 | 54 | 108 | 95 | 304 | 399 |
| Clinical Sensitivity (95% CI) | 72.2% (59.1%- 82.4%) | | | 53.7% (43.1–64.0%) | | |
| Ct ≤20 | 95% (76.4–99.1%), 20 | | | 59.2% (44.2–73.0%), 49 | | |
| Ct ≤25 | 87.9% (72.7–95.2%), 33 | | | 59.0% (47.3–70.0%), 78 | | |
| Ct ≤33 | 76.5% (63.2–86.0%), 51 | | | 56.7% (45.8–67.1%), 90 | | |
| Clinical Specificity (95% CI) | 98.1% (90.2–99.7%) | | | 99.3% (97.6–99.9%) | | |
| Overall Clinical Sensitivity (95% CI) | 60.4% (52.4–67.9%), 149 | | | | | |
| Overall Clinical Specificity (95% CI) | 99.2 (97.6–99.7%), 358 | | | | | |

RT-qPCR = Real-time quantitative polymerase chain reaction, Ct = cycle threshold, CI, confidence interval

**Table 3. Results and clinical sensitivity and specificity of the ActiveXpress+ COVID-19 Complete Testing Kit based on COVID-19 RT-qPCR result in the UK and Peru.**

| Results of ActiveXpress+ | Peru | | | United Kingdom | | |
|---|---|---|---|---|---|---|
| | Confirmed by RT-qPCR | | | | | |
| | Positive | Negative | Total | Positive | Negative | Total |
| Positive | 33 | 0 | 33 | 10 | 1 | 11 |
| Negative | 21 | 66 | 87 | 1 | 199 | 200 |
| Total | 54 | 66 | 120 | 11 | 200 | 211 |
| Clinical Sensitivity (95% CI) | 61.0% (47.8–73.0%) | | | 90.9% (58.7–99.8%) | | |
| Ct ≤20 | 100.0% (74.1–100.0%), 11 | | | 100.0% (15.8–100.0%), 2 | | |
| Ct ≤25 | 100.0% (88.3–100.0%), 29 | | | 100.0% (63.06–100.0%), 8 | | |
| Ct ≤33 | 75.0% (60.6–85.4%), 44 | | | 90.9% (58.72–99.8%), 11 | | |
| Clinical Specificity (95% CI) | 100.0% (94.5–100.0%), 66 | | | 99.5% (97.25–100.0%), 200 | | |
| Overall Clinical Sensitivity (95% CI) | 66.2% (54.0–76.5%), 65 | | | | | |
| Overall Clinical Specificity (95% CI) | 99.6% (97.9–99.9%), 266 | | | | | |

RT-qPCR = Real-time quantitative polymerase chain reaction, Ct = cycle threshold, CI, confidence interval

99.3% [95% CI 97.6–99.9%], respectively, and median of RT-qPCR Ct value of 20.0 [Q1-Q3 15.6–23.8]. In this group the median days from symptom onset was 1 day [Q1-Q3 1–3].

During the enrolment period in the UK for the ActiveXpress+, 11 of 211 (5.2%) specimens were positive for COVID-19 by RT-PCR (see Table 3). Ten of the RT-PCR-positive samples (90.9%) were Ag-RDT positive, while the remaining was Ag-RDT negative. Of the 307 RT-qPCR negative specimens, only 2 were antigen-positive. The sensitivity and specificity for the ActiveXpress+ based on RT-PCR was 90.9% [95% CI 58.7–99.8%] and 99.5% [95% CI 97.3–100.0%], respectively, and median for PCR Ct values was 22.0 [Q1-3 20.5–23.6]. In this group the median days of from symptom onset was 2 days [Q1-Q3 1–3].

Subgroup analyses of the Peruvian and UK evaluation cohorts (Table 4) were performed to determine any associated differences in viral antigen detection/sensitivity compared to vaccination status and days from symptom onset to test. No discernible differences in viral antigen detection/sensitivity were detected for days from symptom onset to test and participants who were vaccinated vs non-vaccinated or those who received 1 dose vs 2 doses of the COVID-9 vaccine in the UK for either evaluation cohorts of the GENEDIA or ActiveXpress+ (all p values >0.05), the subgroup evaluation for Peru´s cohort was not performed due to not availability of vaccines in the country at time of enrolment.

Also the values of Likelihood ratio + and–were calculated, for GENEDIA the overall LR + value was 72, and overall LR–has a value of 0.39; while for ActiveXpress+ the LR + value was 220, and overall LR- has a value of 0.33. In both brands the values of Likelihood ratio were highly relevant to consider the COVID-19 diagnostic if a participant had a positive Ag-RDT result.

### 3.2 Analytical evaluation

The LOD of GENEDIA W and ActiveXpress+ was 5.0 x $10^2$ pfu/ml which equals to approximately 1.0 x $10^4$ gcn/ml fulfilling the recommendations in the WHO Target Product Profile for SARS-CoV-2 Ag-RDT [14].

### 4. Discussion

The study aimed to evaluate the diagnostic performance of GENEDIA and ActiveXpress+ Ag-RDTs. Determining the diagnostic performance of Ag-RDTs is important because this gives

**Table 4. Ag-RDT result by onset of symptoms, and vaccinated individuals in Peru and the UK.**

| Cohort | Peru | | | | United Kingdom | | | |
|---|---|---|---|---|---|---|---|---|
| | Ag-RDT Positive (n, %) | Ag-RDT Negative (n, %) | Sensitivity[a] | 95% CI | Ag-RDT Positive (n, %) | Ag-RDT Negative (n, %) | Sensitivity[a] | 95% CI |
| **GENEDIA** | | | | | | | | |
| **Days from symptom onset** | | | | | | | | |
| Days < 0–3 | 11, 40.7% | 16, 59.3% | 91.7% | 64.6–98.5% | 37, 11.7% | 279, 88.3% | 50.7% | 38.4–63.0% |
| Days 4–7 | 25, 37.9% | 41, 62.1% | 72.7% | 55.8–84.9% | 10, 20.8% | 38, 79.2% | 86.8% | 71.9–95.6% |
| Days 8+ | 3, 21.4% | 11, 78.6% | 44.4% | 18.9–73.3% | 3, 12.5% | 21, 87.5% | 42.9% | 9.9–81.6% |
| **Vaccination received** | | | | | | | | |
| Vaccinated | N/A | N/A | N/A | N/A | 5, 12.5% | 35, 87.5% | 38.5% | 13.9–68.4% |
| Not vaccinated | N/A | N/A | N/A | N/A | 23, 10.2% | 201, 89.7% | 93.0% | 88.6–96.1% |
| Not disclosed | N/A | N/A | N/A | N/A | 25, 18.5% | 110, 81.5% | 80.0% | 71.3–87.0% |
| **ActiveXpress+** | | | | | | | | |
| **Days from symptom onset** | | | | | | | | |
| Days < 0–3 | 9, 29.0% | 22, 71.0% | 64.3% | 38.8–83.7% | 6, 3.7% | 155, 96.3% | 100.0% | 39.8–100.0% |
| Days 4–7 | 22, 29.7% | 52, 70.3% | 61.1% | 44.9–75.2% | 5, 15.6% | 27, 84.4% | 100.0% | 39.8–100.0% |
| Days 8+ | 2, 13.3% | 13, 86.7% | 50.0% | 15.0–85.0% | 0, 0.0% | 10, 100.0% | N/A | N/A |
| **Vaccination received** | | | | | | | | |
| Vaccinated | N/A | N/A | N/A | N/A | 4, 4.4% | 85, 95.5% | 100.0% | 29.2–100.0% |
| Not vaccinated | N/A | N/A | N/A | N/A | 7, 5.9% | 110, 94.0% | 87.5% | 47.4–99.7% |
| Not disclosed | N/A | N/A | N/A | N/A | 0, 0.0% | 5, 100.0% | N/A | N/A |

[a] As compared to RT-qPCR

RT-qPCR = Real-time quantitative polymerase chain reaction, Ct = cycle threshold, CI, confidence interval

an indication of their clinical utility in a pandemic setting, as such tools could improve the time of response and clinical decision making. An adequate sensitivity and specificity values may lead to an increase in testing capacity in low-income settings, particularly where no RT-qPCR can be deployed, and simplifying contact tracing.

A Cochrane systematic review of 48 study cohorts which assessed Ag-RDT performance reported a paucity of high-quality data from prospective studies. Of the evidence identified, sensitivities of between 34.1% (95% CI 29.7% to 38.8%; Coris Bioconcept) to 88.1% (95% CI 84.2% to 91.1%; SD Biosensor STANDARD Q) were reported in symptomatic participants (15). Data of previous field evaluations of Ag-RDT shows variation in sensitivity, in Uganda the STANDARD Q COVID-19 Ag test showed a sensitivity of 70% and specificity of 92% (16), and, in Spain, an evaluation in primary health centers with Panbio™ COVID-19 Ag Rapid Test Device showed sensitivity and specificity values of 79.6% (95%CI 67.0–88.8%) and 100% (95% CI 98.7–100%) respectively (17). Also, an Ag-RDT evaluation in Egypt with SD Biosensor STANDARD Q [18] was made entirely in a laboratory setting and showed sensitivity of 78.2%, the sensitivity values varies from study to study which could be explained by variation in sample collection and epidemiological characteristics of different populations [14–20]. Of many

Ag-RDT performance studies previously carried out, sensitivity values are found to be consistently higher for Ct <25 [15–18].

In this study, the overall clinical sensitivity using GENEDIA (60.4% [95% CI 52.4–67.9]) and ActiveXpress+ (66.2% [95% CI 54.0–76.5]) was lower than reported from manufacturer (GENEDIA: 87.25%, ActiveXpress+: 90%), values which are below the recommended sensitivity for Ag-RDTs according to WHO [8]. Also, GENEDIA brand showed better sensitivity for SARS-CoV-2 detection when RT-qPCR had Ct<25, the overall of UK and Perú cohort for these Ct values was 67.5% and superior if compared with the overall sensitivity for this brand, which is concordant with WHO recommendations for Ag test use [8] and with previous studies [17–20]. Also, for the ActiveXpress+ evaluation, there is a difference between Peru´s (61.0%) and UK sensitivity (90.9%), that could be explained by the number of participants with positive results enrolled for each cohort (54 and 11 in Perú and UK, respectively) due to overestimation of sensitivity in context with low prevalence as UK, as well as the differences in the type of samples used for RT- PCR (NP swabs vs. NP swabs and NT swabs) as the amount of RNA collection is dependent of the site of sample collection and these RNA values are strongly related with sensitivity of PCR assays (19)whereas sample processing of Ag-RDT test was performed in different scenarios. In Peru, Ag-RDTs were undertaken by clinical research nurses, whereas in the UK the tests were run by laboratory scientists. Previous research has suggested that the sensitivity of Ag-RDTs is higher when run by laboratory scientists than when run by healthcare professionals. For example, the Innova Ag-RDT was shown to have a sensitivity of 70% when run by healthcare professionals, versus 79% when run by laboratory scientists [10].

The higher sensitivity with low Ct values is a constant observation between previous evaluations performance of Ag-RDTs in patients with high viral loads and it is relevant as several studies associate high viral loads with worse prognosis and high rates of severity disease [19]. Moreover, these patients are more infectious, and their early detection could facilitate epidemiological surveillance and contact tracing [20].

Interestingly, as the median days from symptom onset increased, the sensitivity of both Ag-RDTs decreased in Peru's cohort. Similar to previous evaluations of Ag-RDTs where sensitivity decreased among symptomatic patients when compared with testing within 5 days and >5 days of symptom onset [21]. However, in the UK cohort of this study, this trend was not observed for either Ag-RDT, with no significant difference in sensitivity at <0–3 days vs 4–7 days from symptom onset found during the ActiveXpress+ evaluation, also there was no specific trend of higher sensitivity values related with days of symptoms in GENEDIA brand for UK cohort. Studies have found the SARS-CoV-2 viral load to increase over the course 5 to 7 days which is generally observed to be the optimal point regarding RT-PCR sensitivity [22]. In this cohort it is likely that the <0–3 days subgroup, did not yet have the viral loads for optimum detection compared to the 4–7 days subgroup.

The circulation of several variants and the prevalence at time of recruitment could affect the observed performance of the Ag-RDT in the two different settings. In Peru, the Alpha (B.1.1.7) variant was first reported in December 2020 [22], the enrolment of participants for GENEDIA started on January 2021 and for Active Xpress+ on February of the same year, during this period, Alpha prevalence was low (1.5%), with the Lambda (C.37) lineage the predominant variant in circulation [23]. Whilst in the UK, Alpha (B.1.1.7) was dominant at the start of the recruitment period in January 2021, with the Delta variant (B.1.617.2) variant rising to a frequency of 46% across the UK by the end of May 2021 [24, 25].

For RT-PCR sample type, the standard of care in the UK recommends combined nose throat swabs for its national COVID-19 testing program. In Peru, the National Institute of Health recommends combined Nasopharyngeal and Oropharyngeal swab for COVID-19

testing. Literature suggests that Ct values and load of SARS-CoV-2 detection in swabs are affected by the collection site, due to high quality and relevant abundance of RNA depends of specific site of collection and this is crucial for the sensitivity of the assays [19]. Regarding type of samples collected by both cohorts in this study, while in Peru only NP swabs were performed for Ag-RDT and RT-PCR, in the UK NP swabs were used for Ag-RDT and nasal and throat swabs for RT-PCR. Reports in late 2020 indicated that nasopharyngeal swabs (NP swabs) were the most specific and accurate swab site for COVID-19 diagnosis, followed by throat swabs [23]. Later research included combined nasal and throat swabs as an additional comparator, showed a higher positivity rate of 100% compared to nasopharyngeal swabs (91.5%) on RT-PCR for SARS-CoV-2 RNA detection [24]. Additionally, the Peru sampling used two NP swabs for Ag-RDT and RT-PCR with unknown comparability between swab types; it is undetermined whether this had an effect on the viral load. With all these considerations, including the difference of sample and RT-qPCR assay type along with the circulation of different variants and differences in recruitment settings between these two cohorts, the variation in sensitivity values of each clinical site would be explained.

Regarding vaccination status, at time of study development, in Perú none of the participants received any doses and data about previously confirmed SARS- CoV-2 infection was not available. The UK cohort included participants who had received vaccination for COVID-19, vaccinated individuals were defined as a person who received any dose of any vaccine brand. When analysed for differences in sensitivity, no significance was detected between the vaccinated and unvaccinated groups. However, an evaluation of effectiveness of COVID-19 vaccines among Health Workers in Israel including the comparison between Ag-RDTs and RT-PCR results, all the participants with positive RT-PCR result undergoes to Ag-RDTs evaluation, were the Ag-RDT positivity for unvaccinated, partially vaccinated and fully vaccinated was 80%, 33% and 21% respectively [25]. Further analysis is warranted to observe whether vaccination affected the viral load detected by Ag-RDTs.

Given the WHO considerations for minimum performance requirements include a sensitivity >80% and specificity values >97% [14], both tests have less-than-optimal performance. Further investigation into the application of these Ag-RDTs across different settings and populations at higher prevalence may provide setting-specific or prevalence-specific performance of SARS-CoV-2 antigen detection and as such further implementation studies are recommended. Edinburgh Genetics have taken Active express off market and are now distributing a new product.

This study had some limitations. First, no follow-up of participants was performed to confirm COVID-19 disease severity, the study was intended to evaluate the usefulness of these diagnostic tool to yield results in the shortest time and with the greatest efficiency, for this reason the specimens were taken at a single time point without perform a second Ag test that may increase assay sensitivity which represents another limitation.

Another limitation is the heterogeneity of participants between both cohorts regarding vaccination status and specimen testing relative to the last vaccine dose. Also, the difference between the rates of positivity on each country at time of study development, while in Peru the study was performed during the peak of the second wave, in UK only a small number of positive samples were included in the UK evaluation owing to the relatively low prevalence in Liverpool during the recruitment period, these differences could lead a bias in the interpretation of results and overestimates sensitivity values in a context where the prevalence was low as UK. In UK cohort, the PCR specimens that had undetermined results were not tested in both platforms due to all the non-paired data were excluded from all the analyses.

Finally, the comparison of select specimens with viral culture to determine the proportion of replication- competent SARS-CoV-2 isolates that tested negative by Ag-RDTs would have

been an important outcome to report, and could be considered as another limitation of this study, we suggest considering this for future performance evaluations related to Ag-RDTs.

To date many evaluations have been restricted to single-centre studies or within one geographical region. This study presents a multicenter evaluation across two different settings with different testing capacity, access to vaccines and prevalence of SARS-CoV-2; in the UK, the introduction of mass vaccination during lowering prevalence, and in Peru, a less controlled increase in positive cases, despite having the world's highest number of deaths per capita at the time of study [26]. With all these considerations, this study is intended to contribute to gather independent data on the performance of COVID-19 Ag-RDTs that are much needed to provide a rapid and accurate detection of SARS-CoV-2 infection to support already stretched healthcare systems especially in LMICs.

In conclusion, our data indicate that the GENEDIA W COVID-19 Ag Device and EDIN-BURGH Genetics ActiveXpress+ COVID-19 have poorer performance to that published by manufacturers for the detection of SARS-CoV-2 from clinical samples.

## Supporting information

**S1 File.**
(XLSX)

## Acknowledgments

LSTM Diagnostics group: Kate Buist, Karina Clerkin, Dr Thomas Edwards, Dr Susan Gould, Caitlin Greenland-Bews, Konstantina Kontogianni, Claudia McKeown, Caitlin R Thompson, Rachel Watkins, Jahanara Wardale, Christopher T Williams and Dominic Wooding. Special thanks to Larysa Mashchenko from the Clinical Research Network in the North-West Coast.

UTB-IMTAvH Group: Tatiana Cáceres, Jordan Bernaldo and Hammerly Lino Fuentes-Rivera.

Condor steering group: Dr A. Joy Allen, Dr Julian Braybrook, Professor Peter Buckle, Professor Paul Dark, Dr Kerrie Davis, Professor Adam Gordon, Ms Anna Halstead, Dr Charlotte Harden, Dr Colette Inkson, Ms Naoko Jones, Dr William Jones, Professor Dan Lasserson, Dr Joseph Lee, Dr Clare Lendrem, Dr Andrew Lewington, Mx Mary Logan, Dr Massimo Micocci, Dr Brian Nicholson, Professor Rafael Perera-Salazar, Mr Graham Prestwich, Dr D. Ashley Price, Dr Charles Reynard, Dr Beverley Riley, Professor John Simpson, Dr Valerie Tate, Dr Philip Turner, Professor Mark Wilcox, Dr Melody Zhifang.

We would like to acknowledge the participants for volunteering to the study and to the CRN for supporting us with the sample collection and recruitment during the study; particularly to Sue Dowling; we also acknowledge the support of the UK National Institute for Health Research Clinical Research Network and the COvid-19 National DiagnOstic Research & evaluation (CONDOR) programme. In Peru, we would like to thank the TB Research Unit of Instituto de Medicina Tropical Alexander Von Humboldt for sample collection and processing.

## Author Contributions

**Conceptualization:** Gail Hayward, Camille Escadafal, Richard Body, Emily R. Adams, Cesar Ugarte-Gil, Ana I. Cubas-Atienzar.

**Data curation:** Margaretha de Vos, Eloïse Cook.

**Formal analysis:** Lorna Finch, Margaretha de Vos.

**Funding acquisition:** Camille Escadafal, Richard Body, Emily R. Adams, Ana I. Cubas-Atienzar.

**Investigation:** Sandra Palomino-Padilla, Lorna Finch, Helen Savage, Luz Villa-Castillo.

**Methodology:** Lorna Finch, Helen Savage.

**Project administration:** Camille Escadafal, Cesar Ugarte-Gil, Ana I. Cubas-Atienzar.

**Resources:** Sandra Palomino-Padilla, Luz Villa-Castillo, Eloïse Cook, Camille Escadafal.

**Software:** Eloïse Cook.

**Supervision:** Sandra Palomino-Padilla, Lorna Finch, Luz Villa-Castillo, Emily R. Adams, Cesar Ugarte-Gil, Ana I. Cubas-Atienzar.

**Writing – original draft:** Sandra Palomino-Padilla, Lorna Finch.

**Writing – review & editing:** Sandra Palomino-Padilla, Margaretha de Vos, Luz Villa-Castillo, Gail Hayward, Camille Escadafal, Richard Body, Emily R. Adams, Cesar Ugarte-Gil, Ana I. Cubas-Atienzar.

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
