## [Decision Letter · Decision Letter 0]

23 Feb 2022

PONE-D-21-39842Diagnostic performance of GENEDIA W and ActiveXpress+ COVID-19 antigens tests among symptomatic individuals in Peru and The United KingdomPLOS ONE

Dear Dr. Finch,

Thank you for submitting your manuscript to PLOS ONE. After careful consideration, we feel that it has merit but does not fully meet PLOS ONE’s publication criteria as it currently stands. Therefore, we invite you to submit a revised version of the manuscript that addresses the points raised during the review process. I agree with the reviewer's overall comments that the manuscript is of interest but requires substantial clarifications in the presentation of the results, as well as significant revisions of the language in the text for linguistic clarity.

We look forward to receiving your revised manuscript.

Kind regards,

Cedric Yansouni, M.D.

Academic Editor

PLOS ONE

2. Please ensure that you have specified (1) whether consent was informed, (2) what type you obtained (for instance, written or verbal, and if verbal, how it was documented and witnessed). If your study included minors, state whether you obtained consent from parents or guardians. If the need for consent was waived by the ethics committee and (3) If you are reporting a retrospective study of medical records or archived samples, please ensure that you have discussed whether all data were fully anonymized before you accessed them and/or whether the IRB or ethics committee waived the requirement for informed consent. If patients provided informed written consent to have data from their medical records used in research, please include this information.

4.Thank you for stating the following in the Acknowledgments Section of your manuscript:

“This work was funded as part of FIND's work as co-convener of the diagnostics pillar of the Access to COVID-19 Tools (ACT) Accelerator, including support from Unitaid [grant number: 2019-32-FIND MDR], the governments of the Netherlands [grant number: MINBUZA-2020.961444] and from UK Department for International Development [grant number 300341-102]. The FALCON study was funded by the National Institute for Health Research, Asthma UK and the British Lung Foundation.”

“This work was funded as part of FIND's work as co-convener of the diagnostics pillar of the Access to COVID-19 Tools (ACT) Accelerator, including support from Unitaid [grant number: 2019-32-FIND MDR], the governments of the Netherlands [grant number: MINBUZA-2020.961444] and from UK Department for International Development [grant number 300341-102]. The FALCON study was funded by the National Institute for Health Research, Asthma UK, and the British Lung Foundation. The funders had no role in study design, data collection and analysis, decision to publish, or preparation of the manuscript.”

6. Please amend your authorship list in your manuscript file to include author Lorna Finch.

Reviewers' comments:

Reviewer's Responses to Questions

**Comments to the Author**

1. Is the manuscript technically sound, and do the data support the conclusions?

Reviewer #1: Partly

2. Has the statistical analysis been performed appropriately and rigorously? 

Reviewer #1: No

3. Have the authors made all data underlying the findings in their manuscript fully available?

Reviewer #1: No

4. Is the manuscript presented in an intelligible fashion and written in standard English?

Reviewer #1: No

5. Review Comments to the Author

Reviewer #1: The authors present a study to evaluate clinical performance of two antigen-based rapid diagnostics tests (Ag-RDTs) for diagnosing SARS-CoV-2 virus: GENEDIA and ActiveXpress+. The study is performed independently in two testing locations- Peru and the United Kingdom. The performance of each of the tests is measured against the RT-qPCR test using sensitivity and specificity as the metrics. The authors conclude that the overall sensitivity for both the tests is lower than what is reported by the manufacturers and lie below the recommended values by WHO. Although the study is important to evaluate the benefits of these tests in controlling the spread of SARS-CoV-2, I am not convinced by the interpretation of some of the data from the study and the corresponding discussion. Additionally, the presentation of key results/comparisons in the current form is difficult to comprehend because of the lack of graphical figures. I recommend that authors should address my following comments in the revision-

1. In lines 204-207, the authors mention that the sensitivity of ActiveXpress+ test in the samples collected at 0-3 days from symptom onset is significantly lower than in the samples collected at 4-7 days from symptom onset for the UK cohort. The sensitivity values presented in the text are extremely low and do not match the data presented in the “sensitivity” column of table 4 for the UK ActiveXpress+ cohort. Additionally, authors mention 161 PCR positive tests for this cohort, which doesn’t match with the data reported in the table 3 (11 PCR positive tests). The authors should address this discrepancy and modify the interpretation of data in the text accordingly.

2. In paragraph 199-211, the authors present the results from the subgroup analysis for the UK cohort but do not compare it (or present) the subgroup analysis for the Peru cohort, especially for days from symptom onset. This should be included in the clinical evaluation section.

3. In lines 264-266, the authors claim that both brands showed better sensitivity when Ct<25 and in the group of participants with <7 days of symptom onset. The authors should clearly specify what these sensitives are being compared to. For example, for the GENEDIA cohort (Table 2), are the individual sensitives for Ct<25 for Peru (87.9%) and UK (59.0%) being compared to the overall sensitivity (60.4%)? If that is the case, the authors’ claim is not valid for the GENEDIA UK Ct<25 cohort. Perhaps, authors should calculate the combined “overall” sensitives for Ct<25 (Peru+UK cohort), which according to my calculation is 67.5%, and compare it to the overall sensitivity (60.4%). On the other hand, if the individual sensitives for Ct<25 for Peru (87.9%) and UK (59.0%) are being compared to the individual clinical sensitivity for Peru (72.2%) and UK (53.7%), an argument should be provided for why such comparison is justified over comparing overall sensitives. Either way, the statistical significance of the comparisons should be provided. Additionally, the same applies to the analysis of sensitives for participants with <7 days of symptom onset.

4. In line 270, it is not clear why the differences in the type of samples used for RT-PCR could be a reason for differences between Peru’s and UK’s sensitives. A clear explanation for this argument should be provided. Furthermore, in line 268, a statistical analysis for significant differences should be provided.

5. In lines 269-270, the authors briefly mention the influence of the low number of RT-PCR positive cases in ActiveXpress+ UK cohort (11) on the sensitivity values. However, the authors should discuss the limitations of this low number and its broad implication on the analysis and the interpretation, especially when making the claims about ActiveXpress+ meeting the minimum sensitivity requirement of WHO in the UK cohort (in line 213).

6. In line 285-287, it is claimed that the decreasing trend of sensitives with the increasing median days from symptom onset is not observed in the UK cohort. It is also claimed that in the UK ActiveXpress+ evaluation, there is a significant difference in the sensitivity at <0-3 days vs 4-7 days. However, this is not supported by the data in table 4 and claims in lines 207-209, as both the sensitives are 100%. Perhaps, the authors intend to make the “significant difference” comparison for the GENEDIA UK cohort. If that is the case, however, lines 210-211 claim that the time from symptom onset has no discernible impact on sensitivity for the GENEDIA UK cohort (p>0.05). Although the first claim in line 285 is still valid based on the data presented in lines 207-211, the language and discrepancy about significant difference between <0-3 days and 4-7 days should be corrected.

7. In paragraph 292-298, the authors discuss that the circulation of several variants at different times during the study could lead to different performances of the tests between the two locations. This suggests that the timeline for the participant recruitment for each test is important for the contextual interpretation of the data. Although the authors mention the overall timeline of the study in each country, it is not clear if the individual timeline for GENEDIA and ActiveXpress+ testing is different or if both the tests were conducted throughout the entire timeline. The authors should specify each of these timelines for both the locations and discuss the performance in context of those timelines.

8. While the discussion is expansive, and probably not all required, the section on limitations is insufficient (paragraph 331-334). This study is limited by:

- Specimens taken at a single time-point; it is, for example, entirely possible that a second Ag test would significantly recuperate assay sensitivity.

- The fact that not every specimen was tested on both platforms. The authors should also explain why this was not done.

- A likely heterogenous group of participants when it comes to vaccination, including vaccine types administered, number of doses, and specimen testing relative to the last vaccine dose.

- While outside the scope of this article, it would have been interesting to compare select specimens with viral culture to determine the proportion of replication-competent SARS-CoV-2 isolates that tested negative by Ag-RDTs. This is another limitation of the study.

9. With the available sensitivity and specificity data, the authors can calculate + and - LR and are encouraged to do so to better contextualize the operating parameters of these assays.

10. How do the authors define vaccinated individuals? Any # of doses? The type of vaccines?

11. What were the median CT values among the vaccinated individuals? Is it possible that vaccinated individuals had, on average, higher CT values? This is important because it could explain the lower sensitivity in this group.

12. Did any of the participants have previously confirmed SARS-CoV-2 infections?

13. In line 183, the authors describe that 212 participants were registered with ActiveXpress+ test in the UK. However, table 3 shows that 211 participants were tested. This discrepancy should be discussed.

Moreover, table 1 also reports 403 patients tested with the GENEDIA, yet table 2 only reports 399 confirmed by RT-PCR. While I recognize that 4 participants did not have paired data, these should be excluded from all analyses.

14. The authors should provide the data (at least in Supplementary Information) from the dilution experiments described in section 2.2 to determine the limit of detection.

15. In addition to the tables, the authors should present the key data in graphical format along with p-values and CI.

In addition to the above, I have minor comments I hope the authors consider:

16. Line 24, the introductory sentence of the abstract is long and would benefit from a grammatical revision.

17. Line 36, similarly, the first two sentences of the results section of the abstract would benefit from a grammatical correction.

18. The grammar should be revised throughout the manuscript. Particular attention should be paid to the following sentences:

- Lines 50, 64, 108

19. The authors are encouraged to have the manuscript reviewed by colleagues who are fluent in the English language to improve the readability of the manuscript.

6. PLOS authors have the option to publish the peer review history of their article (what does this mean?). If published, this will include your full peer review and any attached files.

Reviewer #1: No

---

## [Author Response · Author response to Decision Letter 0]

20 Dec 2022

Dr. Emily Chenette

Editor-in-Chief

PLOS ONE

Dear Dr. Emily Chenette: 

My co-authors and I were pleased to receive your response inviting us to revise and resubmit our manuscript. Accordingly, we would like to submit the enclosed revised paper, “Diagnostic performance of GENEDIA W and ActiveXpress+ COVID-19 antigens tests among symptomatic individuals in Peru and The United Kingdom”, Manuscript PONE-D-21-39842R1, for reconsideration for publication in PLOS ONE Journal.

We express our gratitude for the valuable review of our paper, all the comments have helped us to further strengthen the overall quality of the paper and we have incorporated the suggestions as proposed to improve the manuscript. The specific responses for minor and major comments are listed below in detail:

Regarding journal requirements:

All headings and subheadings have been adjusted to conform to the requirements indicated in PLOS ONE guidelines.

2. Regarding specifications about the type of consent 

We added specifications about informed consent to the Methods section: Study Design, Settings, and Participants

3. We note that the grant information you provided in the ‘Funding Information’ and ‘Financial Disclosure’ sections do not match. When you resubmit, please ensure that you provide the correct grant numbers for the awards you received for your study in the 'Funding Information section. 

Funding information and financial disclosure have been updated

Please remove any funding-related text from the manuscript and let us know how you would like to update your Funding Statement.

Currently, your Funding Statement reads as follows: “This work was funded as part of FIND's work as co-convener of the diagnostics pillar of the Access to COVID-19 Tools (ACT) Accelerator, including support from Unitaid [grant number: 2019-32-FIND MDR], the governments of the Netherlands [grant number: MINBUZA-2020.961444] and from UK Department for International Development [grant number 300341-102]. The FALCON study was funded by the National Institute for Health Research, Asthma UK, and the British Lung Foundation. The funders had no role in study design, data collection, and analysis, decision to publish, or preparation of the manuscript."

Funding-related information has been removed from the manuscript as requested and the statement to include in the online submission is the below

“This work was funded as part of FIND's work as co-convener of the diagnostics pillar of the Access to COVID-19 Tools (ACT) Accelerator, including support from Unitaid [grant number: 2019-32-FIND MDR], the governments of the Netherlands [grant number: MINBUZA-2020.961444] and from UK Department for International Development [grant number 300341-102]. The FALCON study was funded by the National Institute for Health Research, Asthma UK, and the British Lung Foundation. The funders had no role in study design, data collection, and analysis, decision to publish, or preparation of the manuscript." 

Support for this project was provided through funding from The Foundation for Innovative New Diagnostics (FIND)

5. We note that you have indicated that data from this study are available upon request. PLOS only allows data to be available upon request if there are legal or ethical restrictions on sharing data publicly. For more information on unacceptable data access restrictions, please see.

a) If there are ethical or legal restrictions on sharing a de-identified data set, please explain them in detail (e.g., data contain potentially sensitive information, data are owned by a third-party organization, etc.) and who has imposed them (e.g., an ethics

committee). Please also provide contact information for a data access committee, ethics committee, or other institutional body to which data requests may be sent.

There are no legal restrictions on sharing the data set and is available upon request to the investigators

b) If there are no restrictions, please upload the minimal anonymized data set necessary to replicate your study findings as either. Supporting Information files or to a stable, public repository and provide us with the relevant URLs, DOIs, or accession numbers. For a list of acceptable repositories, please see http://journals.plos.org/plosone/s/data-availability#loc-recommended-repositories.

There are no legal restrictions on sharing data sets and the datasets used and/or analyzed during current study are available from the corresponding author on reasonable request.

6. Please amend your authorship list in your manuscript file to include author Lorna Finch.

Author Lorna Finch was added to the manuscript file.

Regarding review comments to the author:

1. In lines 204-207, the authors mention that the sensitivity of the ActiveXpress+ test in the samples collected at 0-3 days from symptom onset is significantly lower than in the samples collected at 4-7 days from symptom onset for the UK cohort. The sensitivity values presented in the text are extremely low and do not match the data presented in the "sensitivity" column of table 4 for the UK ActiveXpress+ cohort. Additionally, the authors mention 161 PCR-positive tests for this cohort, which doesn't match the data reported in table 3 (11 PCR-positive tests). The authors should address this discrepancy and modify the interpretation of data in the text accordingly.

The paragraph was removed because it contained typos that caused a misunderstanding of the data described previously in the tables of the manuscript.

2. In paragraph 199-211, the authors present the results from the subgroup analysis for the UK cohort but do not compare it (or present) the subgroup analysis for the Peru cohort, especially for days from symptom onset. This should be included in the clinical evaluation section.

Peru´s cohort did not have vaccinated population included in the study and for this reason subgroup analysis was not performed, also the information regarding subgroup analysis of the days from symptom onset was removed.

3. In lines 264-266, the authors claim that both brands showed better sensitivity when Ct<25 and in the group of participants with <7 days of symptom onset. The authors should specify what these sensitives are being compared to. For example, for the GENEDIA cohort (Table 2), are the individual sensitives for Ct<25 for Peru (87.9%) and the UK (59.0%) being compared to the overall sensitivity (60.4%)? If that is the case, the authors' claim is not valid for the GENEDIA UK Ct<25 cohort. Perhaps, the authors should calculate the combined "overall" sensitives for Ct<25 (Peru+UK cohort), which according to my calculation is 67.5%, and compare it to the overall sensitivity (60.4%). On the other hand, if the individual sensitives for Ct<25 for Peru (87.9%) and the UK (59.0%) are being compared to the individual clinical sensitivity for Peru (72.2%) and UK (53.7%), an argument should be provided for why such comparison is justified over comparing overall sensitives. Either way, the statistical significance of the comparisons should be provided. Additionally, the same applies to the analysis of sensitives for participants with <7 days of symptom onset. In lines 264-266, the authors claim that both brands showed better sensitivity when Ct<25 and in the group of participants with <7 days of symptom onset. The authors should specify what these sensitives are being compared to. 

Sensitivity values comparison from overall sensitivity (UK + Perú) cohort from Ct <25 and overall sensitivity was stated as suggested. 

4. In line 270, it is not clear why the differences in the type of samples used for RT-PCR could be a reason for differences between Peru’s and UK’s sensitives. A clear explanation for this argument should be provided. Furthermore, in line 268, a statistical analysis for significant differences should be provided.

The explanations of the relation between the collection site and the sensitivity of PCR assays were included in the sentence as suggested; also the statistical analysis for this topic was not added due to lack of relevance for the main objective of the study.

5. In lines 269-270, the authors briefly mention the influence of the low number of RT-PCR positive cases in ActiveXpress+ UK cohort (11) on the sensitivity values. However, the authors should discuss the limitations of this low number and its broad implication on the analysis and the interpretation, especially when making the claims about ActiveXpress+ meeting the minimum sensitivity requirement of WHO in the UK cohort (in line 213).

The overestimation of sensitivities in a low prevalence context was stated in the sentence regarding the low prevalence of COVID cases in the UK cohort during the recruitment period also included in the limitations paragraph as suggested.

6. In line 285-287, it is claimed that the decreasing trend of sensitives with the increasing median days from symptom onset is not observed in the UK cohort. It is also claimed that in the UK ActiveXpress+ evaluation, there is a significant difference in the sensitivity at <0-3 days vs 4-7 days. However, this is not supported by the data in table 4 and claims in lines 207-209, as both the sensitives are 100%. Perhaps, the authors intend to make the “significant difference” comparison for the GENEDIA UK cohort. If that is the case, however, lines 210-211 claim that the time from symptom onset has no discernible impact on sensitivity for the GENEDIA UK cohort (p>0.05). Although the first claim in line 285 is still valid based on the data presented in lines 207-211, the language and discrepancy about significant difference between <0-3 days and 4-7 days should be corrected.

We correct as the reviewer´s observation suggests: "However, in the UK cohort of this study, this trend was not observed for either Ag-RDT, with no significant difference in sensitivity at <0-3 days vs 4-7 days from symptom onset found during the ActiveXpress+ evaluation, also there was no specific trend of higher sensitivity values related with days of symptoms in GENEDIA brand for UK cohort"

7. In paragraph 292-298, the authors discuss that the circulation of several variants at different times during the study could lead to different performances of the tests between the two locations. This suggests that the timeline for the participant recruitment for each test is important for the contextual interpretation of the data. Although the authors mention the overall timeline of the study in each country, it is not clear if the individual timeline for GENEDIA and ActiveXpress+ testing is different or if both the tests were conducted throughout the entire timeline. The authors should specify each of these timelines for both the locations and discuss the performance in context of those timelines.

Specific timelines were added to a better understanding of the context and timeline of the development of this study: “In Peru, the Alpha (B.1.1.7) variant was first reported in December 2020 [22], the enrolment of participants for GENEDIA started on January 2021 and for Active Xpress+ on February of the same year, during this period, the Alpha prevalence was low (1.5%), with the Lambda (C.37) lineage the predominant variant in circulation [23]. Whilst in the UK, Alpha (B.1.1.7) was dominant at the start of the recruitment period in January 2021, with the Delta variant (B.1.617.2) variant rising to a frequency of 46% across the UK by the end of May 2021 [24, 25]".

8. While the discussion is expansive, and probably not all required, the section on limitations is insufficient (paragraph 331-334).

All the suggestions for the limitations section were added to this section due to the high relevance of a broad analysis of these factors in future studies.

9. With the available sensitivity and specificity data, the authors can calculate + and - LR and are encouraged to do so to better contextualize the operating parameters of these assays.

We calculated the LR values as suggested and included them in the results section of the manuscript: ”For GENEDIA the overall LR+ value was 72, and overall LR – has a value of 0.39. For ActiveXpress+ the LR + value was 220, and overall LR- has a value of 0.33”

10. How do the authors define vaccinated individuals? Any # of doses? The type of vaccines?

Specifications about the vaccinated individual’s definition were added: “In the UK, the cohort included participants who had received vaccination for COVID-19, vaccinated individuals were defined as a person who received any dose of any vaccine brand.”

11. What were the median CT values among the vaccinated individuals? Is it possible that vaccinated individuals had, on average, higher CT values? This is important because it could explain the lower sensitivity in this group.

Peru´s cohort does not have any vaccinated participants at the time of recruitment and comparison of CT values between these groups has not been considered.

12. Did any of the participants have previously confirmed SARS-CoV-2 infections?

This data was not available for any of the countries 

13. In line 183, the authors describe that 212 participants were registered with ActiveXpress+ test in the UK. However, table 3 shows that 211 participants were tested. This discrepancy should be discussed. Moreover, table 1 also reports 403 patients tested with the GENEDIA, yet table 2 only reports 399 confirmed by RT-PCR. While I recognize that 4 participants did not have paired data, these should be excluded from all analyses.

This discrepancy was explained in the added paragraph: “In the UK, of the 615 participants recruited, 399 samples were tested with Genedia and 212 with ActiveXpress+ Ag-RDT; 4 participants from Genedia group and one from ActiveXpress+ were PCR undetermined and excluded from all analysis due to not have paired data”.

14. The authors should provide the data (at least in Supplementary Information) from the dilution experiments described in section 2.2 to determine the limit of detection.

The data from the dilution experiments have been included as Supplementary Information.

15. In addition to the tables, the authors should present the key data in graphical format along with p-values and CI.

Although this suggestion has been considered by the research team, after a detailed analysis of the data, we concluded that graphical information is not substantial to the comprehension of the article and does not have any important contribution to this paper.

Regarding minor comments:

16. Line 24, the introductory sentence of the abstract is long and would benefit from a grammatical revision.17. Line 36, similarly, the first two sentences of the results section of the abstract would benefit from a grammatical correction.

18. The grammar should be revised throughout the manuscript. Particular attention should be paid to the following sentences: Lines 50, 64, 108

19. The authors are encouraged to have the manuscript reviewed by colleagues who are fluent in the English language to improve the readability of the manuscript.

Suggestions 16-19 have been assessed and corrected in their respective paragraphs for a better understanding of the readers and reviewers.

We wish to thank you and the reviewers for your insightful comments. These have greatly helped us to improve the quality of our manuscript and we hope that these revisions are sufficient to make our manuscript suitable for publication in the PLOS ONE Journal and look forward to hearing from you at your earliest convenience. 

Yours sincerely,

Sandra Palomino, MD

Instituto de Medicina Tropical Alexander Von Humboldt

Universidad Peruana Cayetano Heredia

Lima, Perú

(+51) 976 810 030 

sandra.palomino.p@upch.pe

---

## [Editor Report · Decision Letter 1]

5 Feb 2023

Diagnostic performance of GENEDIA W and ActiveXpress+ COVID-19 antigens tests among symptomatic individuals in Peru and The United Kingdom

PONE-D-21-39842R1

Dear Dr. Palomino-Padilla,

We’re pleased to inform you that your manuscript has been judged scientifically suitable for publication and will be formally accepted for publication once it meets all outstanding technical requirements.

Kind regards,

Cedric P. Yansouni, M.D.

Academic Editor

PLOS ONE
---

## [Editor Report · Acceptance letter]

22 Feb 2023

PONE-D-21-39842R1 

Diagnostic performance of GENEDIA W and ActiveXpress+ COVID-19 antigens tests among symptomatic individuals in Peru and The United Kingdom 

Dear Dr. Palomino-Padilla:

I'm pleased to inform you that your manuscript has been deemed suitable for publication in PLOS ONE. Congratulations! Your manuscript is now with our production department. 

Kind regards, 

on behalf of

Dr. Cedric P. Yansouni 

Academic Editor

PLOS ONE